# MicroRNA-210 Suppresses NF-κB Signaling in Lipopolysaccharide-Stimulated Dental Pulp Cells Under Hypoxic Conditions

**DOI:** 10.3390/ijms262210837

**Published:** 2025-11-07

**Authors:** Xiyuan Bai, Nobuyuki Kawashima, Shihan Wang, Peifeng Han, Mayuko Fujii, Keisuke Sunada-Nara, Ziniu Yu, Takashi Okiji, Yoshio Yahata

**Affiliations:** 1Department of Pulp Biology and Endodontics, Graduate School of Medical and Dental Sciences, Institute of Science Tokyo, Tokyo 113-8549, Japan; bxy.endo@tmd.ac.jp (X.B.); wang.endo@tmd.ac.jp (S.W.); hpfendo@tmd.ac.jp (P.H.); neigedormir@gmail.com (M.F.); k.nara.endo@tmd.ac.jp (K.S.-N.); yu.endo@tmd.ac.jp (Z.Y.); t.okiji@tky.ndu.ac.jp (T.O.); yahata.yoshio@tmd.ac.jp (Y.Y.); 2Department of Endodontics, The Nippon Dental University School of Life Dentistry at Tokyo, Tokyo 102-8159, Japan

**Keywords:** microRNA-210, hypoxic conditions, human dental pulp cell, NF-κB signaling pathway, proinflammatory cytokine, TGF-beta activated kinase 1 binding protein 1

## Abstract

Dental pulp tissue, enclosed within rigid dentin, is susceptible to bacterial invasion via dentinal tubules, often leading to severe pulpal inflammation. This condition is typically associated with a hypoxic microenvironment, yet the mechanistic link between hypoxia and inflammation remains unclear. We identified a marked upregulation of microRNA-210 (miR-210) in human dental pulp cells (hDPCs) cultured under hypoxic conditions. This study investigated the role of miR-210 in modulating inflammation in lipopolysaccharide (LPS)-stimulated hDPCs. Hypoxic conditions and enforced expression of hypoxia-inducible factor 1α (HIF1α) significantly increased miR-210 levels. While LPS stimulation elevated proinflammatory cytokines (*Interleukin-6*, *Monocyte Chemoattractant Protein-1*, and *Tumor Necrosis Factor Alpha*) and activated nuclear factor-kappa B (NF-κB) signaling, miR-210 overexpression suppressed LPS-mediated cytokine production and NF-κB activity. Luciferase assays revealed that miR-210 targets and negatively regulates TGF-beta activated kinase 1 binding protein 1 (TAB1), a key upstream regulator of NF-κB. Transfection with an miR-210 mimic reduced TAB1 expression, NF-κB activation, and cytokine output in both LPS-stimulated hDPCs and rat pulp tissue ex vivo. Conversely, miR-210 inhibition enhanced TAB1 levels and inflammatory cytokine expression under hypoxic conditions. These findings suggest that miR-210 mitigates inflammation via the TAB1–NF-κB pathway, functioning as a negative feedback regulator. miR-210 may represent a promising therapeutic target for pulpal inflammation.

## 1. Introduction

Dental pulp is a specialized loose connective tissue, derived from the neural crest, that is encased in the rigid dentin matrix and characterized by a rich vascular and neural supply [1]. When bacterial invasion reaches the dental pulp tissue via dentinal tubules as a result of advanced carious lesions or dental trauma, pulpal inflammation is triggered [2]. This response parallels the general systemic inflammatory reactions, functioning as a defense mechanism against bacterial invasion, and is modulated by inflammatory mediators synthesized by the infiltrating inflammatory and immunocompetent cells and resident pulp cells. However, because the pulp is enclosed by dentin, tissue swelling—a hallmark of inflammation—is restricted. Consequently, the intrapulpal pressure rises, leading to ischemia within the pulp tissue [3]. This circulatory disturbance results in localized hypoxic conditions [4,5], which are widely believed to exacerbate pulpal inflammation, thereby perpetuating a vicious cycle of inflammation [6]. In general, hypoxic conditions enhance inflammatory responses, and inflammation exacerbates tissue hypoxic conditions by increasing metabolic demands [7]. This bidirectional relationship forms a positive feedback loop, which has been observed in various pathological conditions, including sepsis, tumor progression, and inflammatory bowel disease [7,8,9].

Various inflammatory diseases, such as rheumatoid arthritis and inflammatory bowel disease, are associated with a hypoxic microenvironment [10,11]. Hypoxia-inducible factor-1α (HIF1α) is a transcription factor activated under hypoxic conditions that stimulates the nuclear factor-kappa B (NF-κB) signaling pathway, which has a central role in the synthesis of proinflammatory mediators [12,13,14]. Upon bacterial invasion of dental pulp, bacterial components such as lipopolysaccharide (LPS) activate Toll-like receptor (TLR) 2/4 and the downstream signaling cascades, particularly the NF-κB signaling pathway [15,16,17,18]. However, the contribution of hypoxic conditions to the progression of inflammation within pulp tissue remains poorly understood.

MicroRNAs (miRNAs) are small, non-coding RNAs 20 to 25 nucleotides in length that post-transcriptionally regulate gene expression and modulate diverse cellular processes [19]. miRNAs exert their effects by binding to complementary sequences within the 3′ untranslated regions (3′-UTRs) of target mRNAs, leading to either mRNA degradation or translational repression [20,21]. Accumulating evidence has highlighted miRNAs as pivotal regulators in the pathogenesis of various human diseases, including cancer, cardiovascular disorders, autoimmune diseases, and inflammatory conditions [22,23,24,25]. Recent reviews have highlighted the pivotal roles of microRNAs in regulating inflammation and disease progression across various pathological conditions [26]. Several inflammation-related miRNAs, such as miR-21, miR-146b, and miR-27a, have been shown to regulate NF-κB signaling and TLR-mediated immune responses in LPS-stimulated human dental pulp cells (hDPCs) [27,28,29].

We screened for miRNAs expressed in the dental pulp under hypoxic conditions by assessing miRNA expression profiles in hDPCs cultured under hypoxic conditions using an miRNA array. miR-210 exhibited the highest upregulation among the differentially expressed miRNAs in hDPCs cultured under hypoxic conditions compared with normoxic conditions (Appendix A). miR-210 is predominantly transcriptionally upregulated under hypoxic conditions by HIF1α and functions as a key mediator of cellular adaptation to hypoxic stress [30]. Notably, miR-210 has been implicated in the regulation of anti-inflammatory responses in acute ischemia [31], macrophage activity [32], and astrocyte function [33]. Accordingly, HIF1α may function as a “double-edged sword,” potentially initiating a negative feedback loop via miR-210 to balance its own proinflammatory effects.

Here, we elucidated the regulatory role of miR-210 in modulating inflammatory cytokine expression in LPS-stimulated hDPCs in vitro and LPS-stimulated rat incisors ex vivo. Additionally, the intracellular signaling pathways governed by miR-210 were systematically evaluated.

## 2. Results

### 2.1. Hypoxic Conditions and HIF1α Overexpression Promoted miR-210 Expression in hDPCs

The expression level of miR-210 in hDPCs was significantly upregulated at 12 h under hypoxic conditions compared with normoxic conditions (*p* < 0.001; Figure 1A). Similarly, enforced expression of HIF1α resulted in significant upregulation of the expression of miR-210 in hDPCs (*p* < 0.01; Figure 1B).

### 2.2. MiR-210 Downregulated the Expression of Proinflammatory Cytokines and the NF-κB Signaling in LPS-Stimulated hDPCs

LPS administration markedly upregulated the mRNA expression of Interleukin-6 (IL6), Monocyte Chemoattractant Protein-1 (MCP1), and Tumor Necrosis Factor Alpha (TNFα) in hDPCs (*p* < 0.0001, Figure 2A). These increases were significantly attenuated by transfection with a miR-210 mimic (*p* < 0.01; Figure 2A). LPS stimulation in hDPCs also elevated the levels of phosphorylated NF-κB p65 (*p* < 0.05; Figure 2B) and NF-κB transcriptional activity (*p* < 0.001; Figure 2C), both of which were downregulated following transfection with a miR-210 mimic (*p* < 0.05; Figure 2B,C).

### 2.3. MiR-210 Targeted TAB1 in NF-κB Signaling

We screened for NF-κB-related factors containing miR-210 binding sites in their 3′-UTRs and identified transforming growth factor-beta-activated kinase 1 binding protein 1 (TAB1), a key mediator of NF-κB signaling, as a candidate. TAB1 mRNA and protein expression was detected in non-stimulated hDPCs, and LPS stimulation did not alter the expression levels (Figure 3A,B). Enforced expression of miR-210 significantly downregulated both the mRNA and protein expression of TAB1 in LPS-stimulated hDPCs (*p* < 0.05 and 0.01, respectively; Figure 3A,B). Reduced TAB1 protein expression was also observed in immunofluorescence assays of LPS-stimulated hDPCs transfected with miR-210 (Figure 3C).

To determine whether miR-210 directly regulates *TAB1* mRNA expression, we conducted a luciferase reporter assay using two pMIR-REPORT vectors containing the two putative miR-210 binding sequences within the TAB1 3′-UTR. Luciferase activity in hDPCs transfected with the reporter vectors was significantly suppressed by enforced expression of miR-210 (*p* < 0.001, Figure 3D). Moreover, the suppressive effect of miR-210 was abolished when the putative miR-210 binding sites were mutated (*p* < 0.01, Figure 3D).

### 2.4. MiR-210 Inhibitor Upregulated the Expression of Proinflammatory Cytokines and NF-κB Signaling in LPS-Stimulated hDPCs

Our results demonstrated that miR-210 is induced by hypoxic conditions and forced HIF1α expression, and miR-210 suppresses NF-κB signaling in LPS-stimulated hDPCs, thereby decreasing inflammatory cytokine expression. Additionally, we found that TAB1 plays a critical role in NF-κB signaling and the production of inflammatory cytokines in LPS-stimulated human dental pulp cells. To further investigate the function of miR-210, we inhibited its expression in hDPCs under hypoxic conditions by transfecting a miR-210 inhibitor and assessed the responsiveness to LPS stimulation. The mRNA expressions of IL6 and MCP1 were significantly upregulated in LPS-stimulated hDPCs under hypoxic conditions (*p* < 0.001), and transfection of the miR-210 inhibitor further increased their mRNA expression levels (*p* < 0.01, Figure 4A). Similarly, the expressions of phosphorylated NF-κB p65 and TAB1 were elevated in LPS-stimulated hDPCs cultured under hypoxic conditions (*p* < 0.001 and *p* < 0.01, respectively) and further increased following miR-210 inhibition (*p* < 0.01 and *p* < 0.05, respectively) (Figure 4B,C). Notably, TAB1 expression was significantly upregulated by the miR-210 inhibitor in hDPCs even without LPS stimulation (*p* < 0.05, Figure 4C).

### 2.5. MiR-210 Downregulated the Expression of Proinflammatory Cytokines and the NF-κB Signaling in LPS-Stimulated Rat Pulp Tissue Ex Vivo

Overexpression of miR-210 (Figure 5A) significantly suppressed the mRNA expression of Il6, Mcp1, and Tnfa in LPS-stimulated rat pulp tissues ex vivo (*p* < 0.05 to 0.001; Figure 5B). miR-210 overexpression also reduced the phosphorylation of NF-κB p65 (*p* < 0.05; Figure 5C). Both the mRNA and protein levels of Tab1 were markedly decreased following miR-210 overexpression (*p* < 0.05 and 0.01; Figure 5D,E). The suppressive effects of miR-210 on Tab1 expression were further validated ex vivo through immunofluorescence staining (Figure 5F).

## 3. Discussion

This study demonstrates that miR-210 negatively regulates pulpal inflammation under hypoxic conditions by suppressing the TAB1–NF-κB signaling pathway. Using both LPS-stimulated hDPCs and LPS-applied rat pulp tissues, we found that miR-210 overexpression significantly reduced the mRNA levels of IL6, MCP1, and TNFα in hDPCs (Figure 2A) and *Il6*, *Mcp1*, and *Tnfα* expression in rat pulp tissue ex vivo (Figure 5B). In contrast, inhibition of miR-210 enhanced *IL6* and *MCP1* expression in hypoxia-exposed hDPCs (Figure 4A), reinforcing the role of miR-210 in dampening inflammatory cytokine synthesis.

Hypoxic conditions activate NF-κB signaling via HIF1α [34], a key transcriptional regulator at the intersection of oxygen deprivation and inflammation. HIF1α promotes NF-κB-dependent cytokine expression in LPS-stimulated macrophages [35] and hypoxic neutrophils [36]. Our findings reveal that HIF1α upregulates miR-210, which in turn attenuates NF-κB activation, forming a negative feedback loop in both hDPCs and rat pulp tissues. miR-210 has been shown to attenuate cardiac apoptosis following myocardial infarction [37], modulate the differentiation of T cell subsets in psoriasis [38], and suppress proinflammatory cytokine production in Leishmania-infected macrophages [32]. As a hypoxia-responsive miRNA, miR-210 contributes to both inflammatory resolution and tissue repair during ischemic injury [31]. Its induction in ischemic pulp tissue suggests a similar dual role in mitigating inflammation and promoting healing.

NF-κB, particularly the p65 subunit, is a central regulator of proinflammatory cytokine transcription [39,40]. Previous studies demonstrated that miR-146b, miR-21, and miR-27a suppress NF-κB-mediated cytokine production in hDPCs [27,28,29]. Our results showed that miR-210 overexpression reduced NF-κB reporter activity (Figure 2C) and p65 phosphorylation in LPS-stimulated hDPCs and rat pulp tissues (Figure 2B and Figure 5C), whereas miR-210 inhibition enhanced p65 phosphorylation (Figure 4B). These results support the role of miR-210 as a key negative regulator of NF-κB signaling, with its inhibitory effect contributing to the downregulation of inflammatory cytokine production. Given that dental pulp tissue exhibits lower oxygen levels than other tissues, even under physiological conditions [41], miR-210 may be constitutively expressed in healthy pulp. Indeed, even in the absence of LPS stimulation, inhibition of miR-210 significantly upregulated TAB1 expression in hDPCs cultured under hypoxic conditions (Figure 4C). Collectively, these results indicate that miR-210 may hold clinical potential as an anti-inflammatory therapeutic target and may play an essential role in maintaining dental pulp homeostasis by modulating basal inflammatory tone and preventing excessive immune activation.

We identified direct binding sites for miR-210 in the TAB1 3′-UTR (Figure 3D). Overexpression of miR-210 significantly suppressed TAB1 expression in hDPCs (Figure 3A–C) and rat pulp tissue ex vivo (Figure 5D–F), whereas its inhibition increased TAB1 expression (Figure 4C). TAB1 is a key component of the NF-κB signaling cascade, activating transforming growth factor-beta-activated kinase 1 (TAK1), which phosphorylates IκB kinase (IKK), leading to IκBα degradation and NF-κB activation [42]. The TAK1–TAB1 complex contributes to inflammatory pathogenesis [43,44], and TAB1 suppression reduces NF-κB activation and cytokine production [28,43,45,46]. Beyond NF-κB regulation, TAB1 mediates p38-mitogen-activated protein kinase (MAPK) activation via autophosphorylation, contributing to myocardial ischemia injury [47], while its knockdown mitigated cardiomyocyte hypertrophy [48]. Our findings provide novel insights into the regulatory function of miR-210 in inflammation, suggesting that miR-210 attenuates inflammatory responses by targeting the TAB1–NF-κB signaling pathway (Figure 6).

This study provides evidence for the therapeutic potential of miR-210 in pulpal inflammation, as evidenced by both in vitro and ex vivo findings. However, the absence of in vivo validation in human pulp tissue is a notable study limitation. Further investigations are required to establish the efficacy, safety, and translational relevance of miR-210 as a candidate for clinical intervention in pulpal inflammatory disorders.

## 4. Materials and Methods

### 4.1. Cell Culture

The study received approval from the Ethical Committee of Tokyo Medical and Dental University (currently Institute of Science Tokyo, Tokyo, Japan; D2023-066). hDPCs obtained from healthy human wisdom teeth were cultured in α minimum essential medium (FUJIFILM Wako Pure Chemical, Osaka, Japan) supplemented with 10% fetal bovine serum (Biowest, Nuaillé, France) and 1% penicillin/streptomycin (FUJIFILM Wako Pure Chemical) at 37 °C in an atmosphere of 5% CO_2_ in air (referred to as 21% O_2_).

Hypoxic conditions (1% O_2_) were established using a gas mixture composed of 1% O_2_, 5% CO_2_, and 94% N_2_. Cells were incubated under hypoxic conditions for 12 h prior to the application of LPS (100 ng/mL; *Escherichia coli* O111:B4, Sigma-Aldrich, St. Louis, MO, USA).

### 4.2. miRNA Array

Total RNA was isolated from hDPCs cultured in 6-well plates (2  ×  10^5^ cells per well) under normoxic or hypoxic conditions (n = 3 per condition), using the mirVana™ miRNA Isolation Kit (Thermo Fisher Scientific, Waltham, MA, USA). RNA integrity was evaluated with a 2100 Bioanalyzer (Agilent, Santa Clara, CA, USA). For miRNA expression profiling, 1 µg of total RNA from both hDPCs cultured under normoxic and hypoxic conditions was analyzed using the Affymetrix GeneChip miRNA 4.0 Array (Thermo Fisher Scientific), and data were processed using the Microarray Data Analysis Tool version 3.2 (Filgen, Nagoya, Japan).

### 4.3. MiR-210 Mimic and Inhibitor Transfection

hDPCs were transfected with the mirVana miRNA mimic for hsa-miR-210 (miR-210 mimic; Thermo Fisher Scientific, Waltham, MA, USA) using Lipofectamine RNAiMAX transfection reagent (Thermo Fisher Scientific). The mirVana miRNA mimic Negative Control #1 (NC; Thermo Fisher Scientific) was used as a negative control. For inhibition experiments, the mirVana miRNA inhibitor miR-210 (miR-210 inhibitor; Thermo Fisher Scientific) was transfected into hDPCs using Lipofectamine RNAiMAX transfection reagent, with the mirVana miRNA inhibitor Negative Control #1 serving as the control.

### 4.4. Ex Vivo miR-210 Mimic Transfection

Animal experiments were approved by the Animal Experimentation Committee of the university (A2023-196C4). Male Sprague-Dawley rats (6 weeks old; CLEA Japan, Tokyo, Japan) were euthanized by CO_2_ inhalation, and the pulp tissues were isolated from maxillary and mandibular incisors. The extracted pulp tissues were transfected ex vivo with mirVana miRNA mimic miR-210 or mirVana miRNA mimic Negative Control #1 using Lipofectamine RNAiMAX. After 24 h, tissues were treated with 200 ng/mL LPS for 3 h.

### 4.5. Reverse Transcription-Quantitative Polymerase Chain Reaction (RT-qPCR)

For mRNA analysis, total RNA was extracted using the QuickGene-Mini80 nucleic acid isolation system (Kurabo, Osaka, Japan). Complementary DNA (cDNA) was synthesized using reverse transcriptase (RevertAid H Minus Reverse Transcriptase, Thermo Fisher Scientific) and PrimeScript™ RT Master Mix (Takara Bio, Kusatsu, Japan). mRNA expression levels were analyzed by a real-time qPCR system (CFX 96, Bio-Rad, Hercules, CA, USA). For miR-210 analysis, total RNA was isolated using the mirVana miRNA Isolation Kit (Thermo Fisher Scientific). cDNA synthesis was performed using miR-210 and U6-specific RT primers in the TaqMan microRNA Assays (Thermo Fisher Scientific) with a microRNA Reverse Transcription Kit (Thermo Fisher Scientific). Real-time qPCR was carried out using the CFX96 system and TaqMan Universal Master Mix II, no UNG (Thermo Fisher Scientific). The formula 2^−ΔΔCt^ was used to calculate relative gene expression, with *ACTB* mRNA or *U6* snRNA serving as internal controls. The primer sequences are listed in Table 1.

### 4.6. Western Blotting

Proteins were isolated with a radioimmunoprecipitation assay (RIPA) buffer supplemented with protease (cOmplete, Sigma-Aldrich) and phosphatase (PhosSTOP, Sigma-Aldrich) inhibitors. Samples were mixed with loading buffer and denatured at 95 °C for 3 min, separated by SDS–polyacrylamide gel electrophoresis (e-PAGEL; ATTO, Tokyo, Japan), and transferred to polyvinylidene difluoride membranes (Immobilon-P; Merck Millipore, Burlington, MA, USA). The membranes were subsequently incubated with primary antibodies, including rabbit anti-TAB1 (1:300, Q15750, polyclonal, rabbit; Proteintech, Rosemont, IL, USA), rabbit anti-NF-κB p65 (1:1000, D14E12, monoclonal, rabbit; Cell Signaling Technology, Danvers, MA, USA), rabbit anti-phospho-NF-κB p65 (1:1000, Ser536, 93H1, monoclonal; Cell Signaling Technology), and anti-glyceraldehyde-3-phosphate dehydrogenase (GAPDH, 1:4000, PM053-7; Medical and Biological Laboratories, Nagoya, Japan). Horseradish peroxidase (HRP)-conjugated anti-rabbit IgG (1:5000, W4011; Promega, Madison, WI, USA) was used as the secondary antibody. Protein bands were visualized using a chemiluminescent HRP substrate (Immobilon, Merck Millipore), and images were obtained with the LAS-3000 mini-imaging system (FUJIFILM Wako Pure Chemical). Densitometric analysis was performed using ImageJ software (version 2; National Institutes of Health, Bethesda, MD, USA) to quantify band intensity and determine the relative ratios.

### 4.7. HΙF1α Expression Vector 

To induce transient expression of HIF1α, a mammalian expression vector containing the cytomegalovirus promoter and encoding an oxygen-insensitive mutant form of HIF-1α (HA-HIF1α-P402A/P564A mutant-pcDNA3, a gift from William Kaelin, Addgene plasmid #18955; https://www.addgene.org/18955/; accessed on 18 June 2018; RRID: Addgene_18955, Watertown, MA, USA [49]) was used. This vector contains double point mutations in HIF-1α (P402A and P564A) that inhibit degradation and promote the transcriptional activity of HIF-1α. As a mock control, an enhanced green fluorescent protein (EGFP) expression vector (pMAX-EGFP, Lonza, Gampel, Switzerland) was used. Vectors were transfected into hDPCs using Lipofectamine LTX with Plus reagent (Thermo Fisher Scientific) following the manufacturer’s protocol.

### 4.8. Luciferase Assays

The NF-κB luciferase assay was conducted using the pGL4.32 (luc2P/NF-κB-RE/Hygro) reporter plasmid (Promega, Madison, WI, USA), which carries five tandem NF-κB response elements. The reporter plasmid together with either the miR-210 mimic or negative control was transfected into hDPCs using Lipofectamine 3000 transfection reagent (Thermo Fisher Scientific), followed by stimulation with LPS (100 ng/mL). For the TAB1 3′-UTR luciferase assay, the synthesized TAB1 3′-UTR containing either the wild-type or mutated hsa-miR-210-3p target sequences (400 bp each; Eurofins Genomics, Ebersberg, Germany) were inserted into the XhoI and HindIII sites of the pMIR-REPORT vector (Thermo Fisher Scientific). The reporter vectors were co-transfected into hDPCs along with the miR-210 mimic or NC using Lipofectamine 3000 transfection reagent. Transfected cells were lysed using a luciferase cell culture lysis reagent, and luciferase activity was measured with a luciferase assay system (Promega) and luminometer (Luminescence PSN; ATTO).

### 4.9. Immunofluorescence

Following PBS washing, hDPCs were fixed in 4% paraformaldehyde at 4 °C for 10 min. Pulp tissues were fixed in 4% paraformaldehyde at 4 °C overnight, embedded in an embedding medium (Tissue-Tek O.C.T. Compound, Sakura Finetek, Tokyo, Japan), and sectioned into 10 µm slices using a cryostat (Leica CM1860, Leica Biosystems, Nussloch, Germany). Fixed hDPCs and rat pulp tissue sections were incubated with a primary rabbit anti-ΤAΒ1 antibody (1:300, Q15750) at 4 °C overnight. After washing, samples were incubated with Alexa Fluor 488-conjugated donkey anti-rabbit IgG (1:500, ab150077, Abcam, Cambridge, UK) and mounted using Fluoroshield Mounting Medium with 4′,6-diamidino-2-phenylindole (ab10413; Abcam). PBS was used as a negative control. Fluorescence images were acquired using a confocal laser scanning microscope (Leica TCS-SP8; Leica Microsystems, Wetzlar, Germany).

### 4.10. Statistical Analysis

All experiments were performed with three independent biological replicates. Data are presented as the mean and standard deviation (SD). Statistical analysis was conducted using GraphPad Prism 9.0 (GraphPad Software, San Diego, CA, USA). For comparisons among three or more groups, data were analyzed by one-way ANOVA followed by Tukey–Kramer or Bonferroni post hoc tests, while comparisons between two groups were performed using Student’s *t*-test. *p* < 0.05 indicated statistical significance.

## 5. Conclusions

miR-210, which was upregulated under hypoxic conditions, attenuated the expression of *IL6*, *MCP1* and *TNFα* in LPS-stimulated hDPCs by modulating NF-κB signaling through direct targeting of TAB1. These findings underscore the regulatory role of miR-210 in pulpal inflammation, particularly within hypoxic microenvironments, and suggest its potential as a molecular modulator of inflammation in dental pulp tissues.

## Figures and Tables

**Figure 1 ijms-26-10837-f001:**
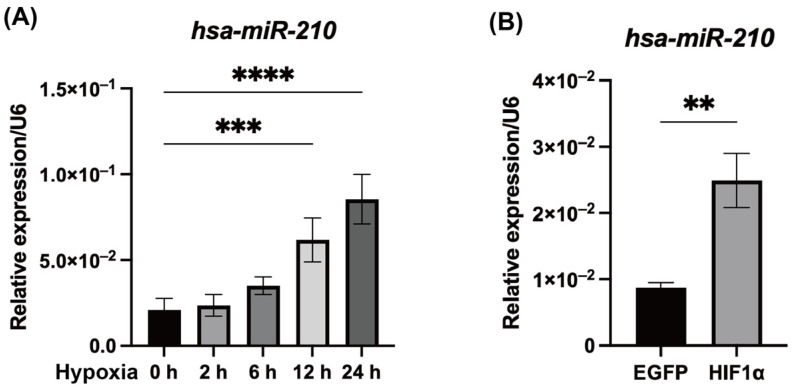
Hypoxic conditions and HIF1α overexpression promoted miR-210 expression in hDPCs. (**A**) Hypoxic conditions and (**B**) enforced expression of HIF1α significantly increased the expression of miR-210 in hDPCs. Data are presented as mean ± SD (n = 3). ** *p* < 0.01, *** *p* < 0.001; **** *p* < 0.0001. EGFP (control): enhanced green fluorescent protein expression vector.

**Figure 2 ijms-26-10837-f002:**
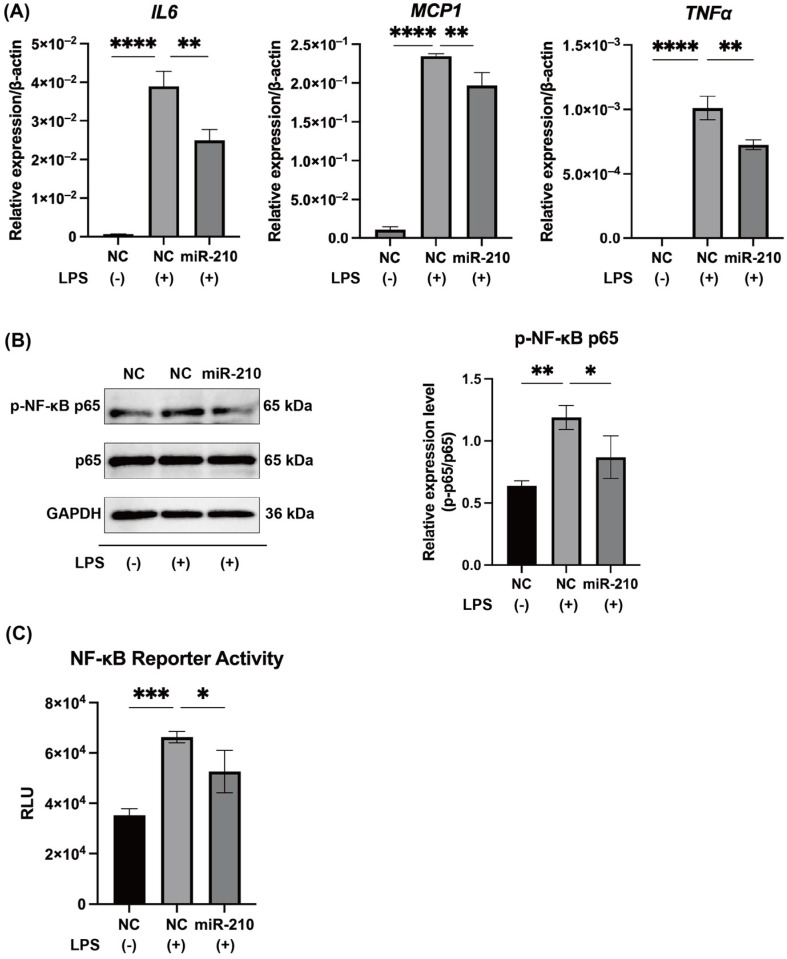
MiR-210 downregulated the mRNA expression of proinflammatory cytokines and NF-κB signaling induced by LPS. (**A**) Enforced expression of miR-210 significantly reduced the mRNA expression of IL6, MCP1, and TNFα induced by LPS. (**B**) Phosphorylation of NF-κB p65 and (**C**) NF-κB transcriptional activity induced by LPS were also suppressed by miR-210 overexpression. Data are presented as mean ± SD (n = 3). * *p* < 0.05; ** *p* < 0.01; *** *p* < 0.001; **** *p* < 0.0001.

**Figure 3 ijms-26-10837-f003:**
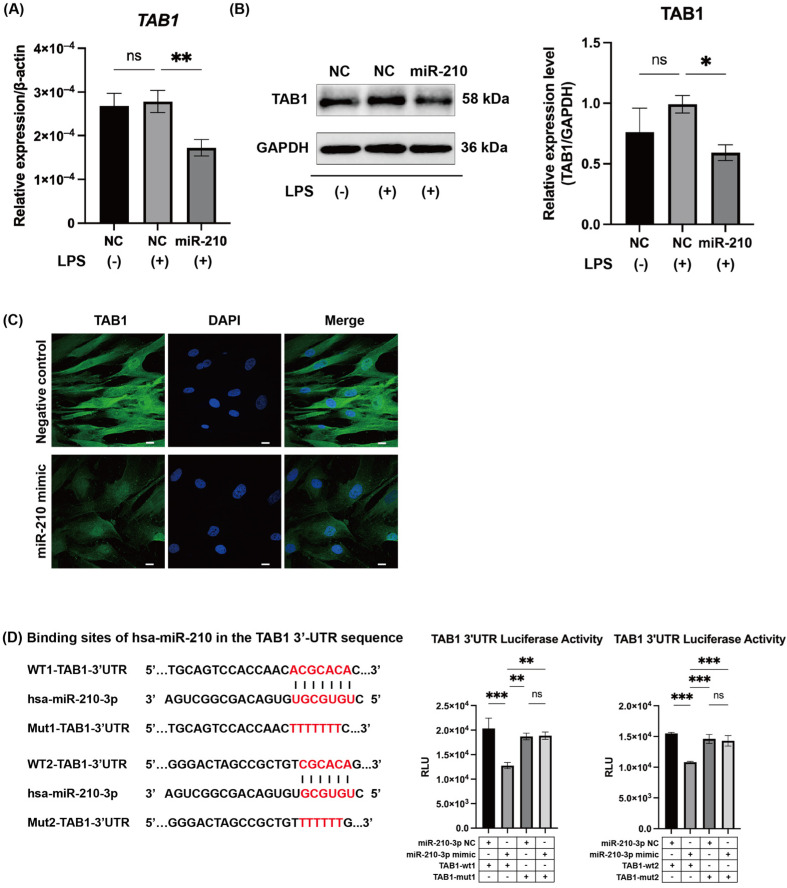
MiR-210 mimic downregulated the expression of TAB1 in LPS-stimulated hDPCs. (**A**) TAB1 mRNA and (**B**) protein expressions were significantly downregulated by transfection of miR-210 mimic in LPS-stimulated hDPCs (mean ± SD, n = 3). (**C**) Immunofluorescence staining revealed reduced expression of TAB1 in miR-210 mimic-transfected and LPS-stimulated hDPCs. Scale bars = 10 µm. (**D**) Two putative miR-210 binding sites were identified in the 3′-UTR of TAB1. Luciferase activity in hDPCs transfected with pMIR-REPORT vectors containing TAB1 3′-UTR sequences containing the putative miR-210 binding sites was significantly suppressed by miR-210 mimic transfection. This suppressive effect was abolished when the miR-210 binding sites were mutated (mean ± SD, n = 3). * *p* < 0.05; ** *p* < 0.01; *** *p* < 0.001; ^ns^ *p* ≥ 0.05. NC: miRNA mimic Negative Control #1; miR-210: miRNA mimic for miR-210.

**Figure 4 ijms-26-10837-f004:**
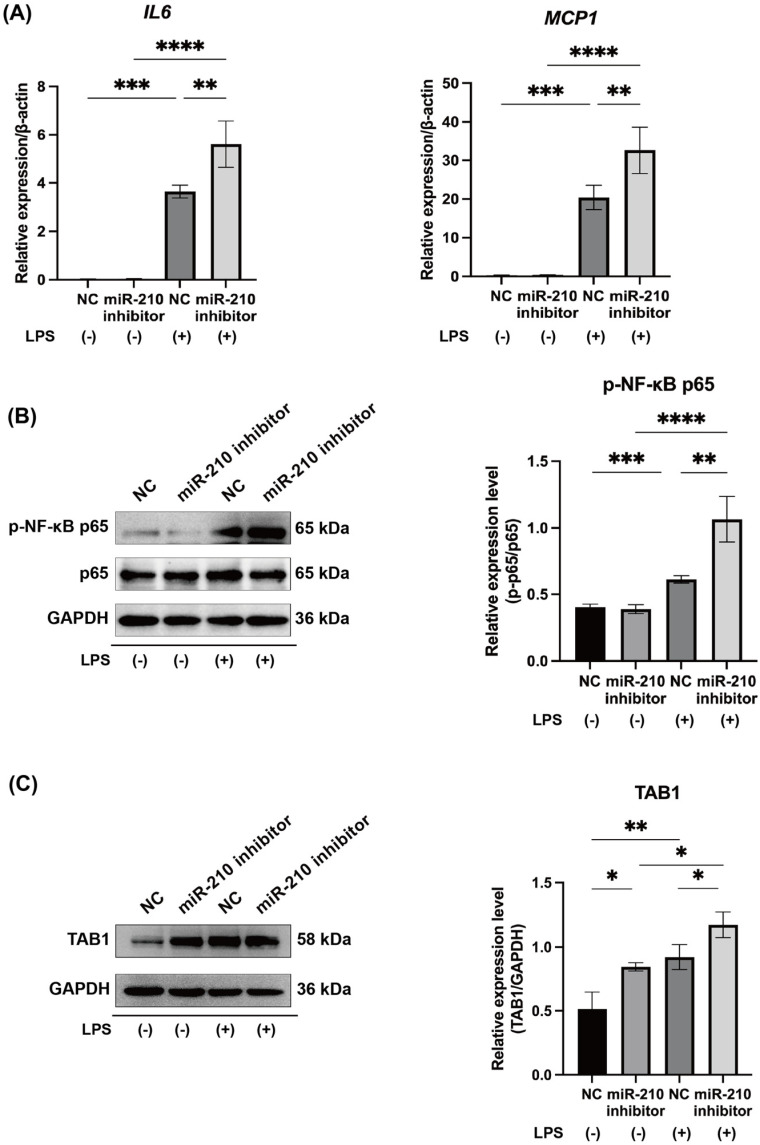
MiR-210 inhibitor promoted the expression of proinflammatory cytokines and TAB1 in LPS-stimulated hDPCs under hypoxic conditions. (**A**) LPS stimulation significantly upregulated *IL6* and *MCP1* mRNA expression (n = 3) and (**B**) phosphorylated NF-κB p65 level (n = 3) in hDPCs cultured under hypoxic conditions. Transfection with a miR-210 inhibitor further enhanced these responses. (**C**) Under hypoxic conditions, TAB1 expression was minimal in hDPCs, but markedly increased following LPS stimulation. miR-210 inhibition further promoted TAB1 expression (n = 3). * *p* < 0.05; ** *p* < 0.01; *** *p* < 0.001; **** *p* < 0.0001. NC (**A**–**C**): miRNA inhibitor Negative Control #1; miR-210 inhibitor: miRNA inhibitor for miR-210.

**Figure 5 ijms-26-10837-f005:**
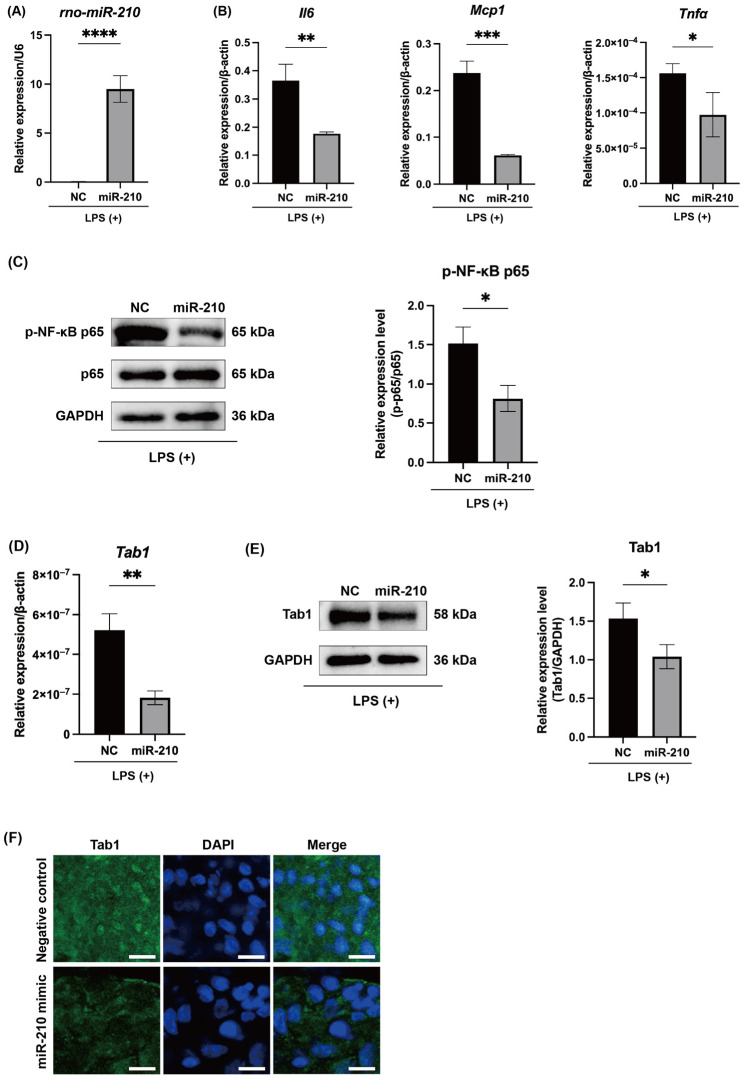
MiR-210 mimic reduced proinflammatory mediators, NF-κB signaling, and Tab1 expression in LPS-applied rat pulp tissues ex vivo. (**A**) Upregulation of miR-210 was confirmed following miR-210 mimic transfection in LPS-stimulated rat pulp tissues. Enforced expression of miR-210 significantly downregulated (**B**) the mRNA expression of Il6, Mcp1 and Tnfα (mean ± SD, n = 3), (**C**) phosphorylated NF-κB p65 levels, and (**D**) mRNA and (**E**) protein expression of Tab1. (**F**) Immunofluorescence staining confirmed reduced Tab1 expression in miR-210 mimic-transfected, LPS-stimulated rat pulp tissue. Scale bars = 5 µm. * *p* < 0.05; ** *p* < 0.01; *** *p* < 0.001; **** *p* < 0.0001. NC: miRNA mimic Negative Control #1; miR-210: miRNA mimic for miR-210.

**Figure 6 ijms-26-10837-f006:**
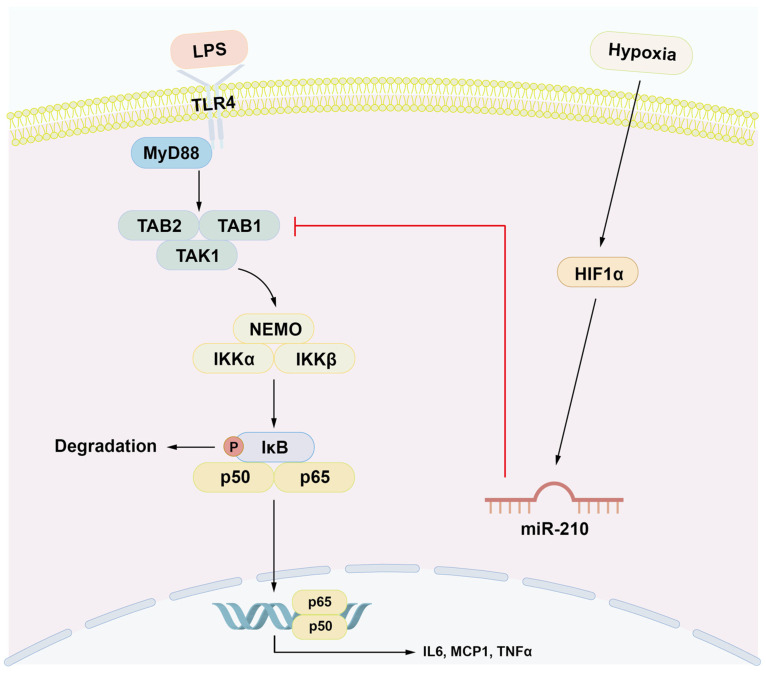
A schematic diagram of the proposed regulatory role of miR-210 in hDPCs. Upon binding of LPS to TLR4, the NF-κB signaling pathway is activated, leading to the production of proinflammatory mediators such as *IL6*, *MCP1*, and *TNFα*. HIF1α upregulates miR-210 in hDPCs under hypoxic conditions. miR-210 in turn acts as a negative regulator of the NF-κB signaling pathway by targeting TAB1.

**Table 1 ijms-26-10837-t001:** Sequences of primers used for RT-qPCR.

Gene	Forward	Reverse	Accession No.	Size, Bp
<human>				
*ACTB*	5′-GTAGCACAGCTTCTCCTTAATGTCA-3′	5′-CTGACTGACTACCTCATGAAGATCC-3′	NM_001101.3	102
*IL6*	5′-TATACCTCAAACTCCAAAAGACCAG-3′	5′-ACAAGAGTAACATGTGTGAAAGCAG-3′	NM_000600.4	157
*MCP1*	5′-CACCTGCTGTTATAACTTCACCAAT-3′	5′-GTTGAAAGATGATAAGCCCACTCTA-3′	NM_002982.4	130
*TNFα*	5′-CCTGGTATGAGCCCATCTATCTG-3′	5′-GCAATGATCCCAAAGTAGACCTG-3′	NM_000594.3	130
*TAB1*	5′-ATCCCTCAGTGCCAACTAAACC-3′	5′-GAAGATCCCAGTGCACAAGTCA-3′	NM_153497.3	137
<rat>				
*Actb*	5′-GTAAAGACCTCTATGCCAACACAGT-3′	5′-GGAGCAATGATCTTGATCTTCATGG -3′	NM_031144.3	127
*Il6*	5′-TAAGGACCAAGACCATCCAACTCAT-3′	5′-AGTGAGGAATGTCCACAAACTGATA-3′	NM_012589.2	125
*Mcp1*	5′-CTAAGGACTTCAGCACCTTTGAATG-3′	5′-GTTCTCTGTCATACTGGTCACTTCT-3′	NM_031530.1	120
*Tnfα*	5′-AAACGGAGCTAAACTACCAGCTATC-3′	5′-CCTGGTCACCAAATCAGCATTATTA-3′	NM_012675.3	139
*Tab1*	5′-TAGTGTCTGCTTCTGTTAGATCCTG-3′	5′-AATCAGCTTCCTCATCAGAGTGAAA-3′	NM_001109976.2	134

Abbreviations: ACTB: actin beta; IL6: interleukin 6; MCP1: monocyte chemotactic protein 1; TNFα: tumor necrosis factor alpha; TAB1: transforming growth factor-beta-activated kinase 1 binding protein 1.

## Data Availability

The original contributions presented in this study are included in the article/Appendix A. Further inquiries can be directed to the corresponding author.

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
