# Peer review of "MicroRNA-210 Suppresses NF-κB Signaling in Lipopolysaccharide-Stimulated Dental Pulp Cells Under Hypoxic Conditions"

_ijms, 2025, doi:10.3390/ijms262210837_

Round 1

Reviewer 1 Report

Comments and Suggestions for Authors

Comments:

This study investigates the role of microRNA-210 in hypoxic pulpitis. The findings demonstrate that hypoxia and HIF1α significantly induce miR-210 expression. In both LPS-stimulated human dental pulp cells and rat ex vivo pulp tissue, miR-210 suppresses the production of pro-inflammatory cytokines (IL-6, MCP-1, TNF-α) by directly targeting TAB1 and inhibiting the NF-κB signaling pathway. These results reveal miR-210's crucial function as a negative feedback regulator in the hypoxic pulp microenvironment, highlighting its potential therapeutic value. However, the following issues need to be addressed before consideration for publication.

Comments:

  1. Page 2 notes HIF1α's pro-inflammatory role (line 57) and its induction of miR-210 (line 76). To better frame the study, explicitly state that HIF1α may function as a "double-edged sword," potentially initiating a negative feedback loop via miR-210 to balance its own pro-inflammatory effects. This provides stronger rationale for investigating the miR-210/NF-κB axis.
  2. On page 4, Figure 2, the results show that the miR-210 mimic significantly inhibits pro-inflammatory factors and the NF-κB pathway. However, it was not verified whether the miR-210 mimic itself affects cell viability or baseline inflammation levels.
  3. The Discussion section is overly verbose and complex and should be streamlined.
  4. On page 9, line 209, the physiological significance of miR-210 is pointed out. It is recommended to connect this with the experimental result on page 6, line 152: "Even without LPS stimulation, the miR-210 inhibitor significantly upregulates TAB1 expression in hDPCs," to mutually corroborate and strengthen the argument.
  5. On page 9, line 209, the physiological significance of miR-210 is pointed out. It is recommended to connect this with the experimental result on page 6, line 152: "Even without LPS stimulation, the miR-210 inhibitor significantly upregulates TAB1 expression in hDPCs," to mutually corroborate and strengthen the argument.
  6. The term "n=3" is used several times in the manuscript, but it is not clarified whether this refers to technical replicates (measured three times consecutively) or biological replicates. It is recommended to clarify this in the Methods section.
  7. In the Conclusion section, the statement "miR-210 attenuates... by directly targeting TAB1 and modulating NF-κB signaling" is presented as a definitive conclusion derived from the experiments. The wording is overly absolute; it is recommended to modify the phrasing to reflect a more nuanced interpretation based on the data.
  8. Large blank sections appear on pages 3 and 7, suggesting formatting or conversion issues. It is recommended to revise the typesetting.

Reviewer 2 Report

Comments and Suggestions for Authors

The work by Bai et al., discovered that HIF1α upregulates miR-210, which in turn attenuates NF-κB activation, forming a negative feedback loop, in both hDPCs and rat pulp  tissues. Overall, the study is interesting and well-written. I have prepared a few suggestions that may help the authors in providing a revised version of their work for further consideration.

  1. The authors correctly acknowledge the lack of experiments in human pulp tissue. Could data from publicly available studies help in that direction? For example PMID: 22595106 or other microarray or RNA-seq analysis in human pulp under inflammatory conditions. Are your main differentially expressed miRNAs also differentially expressed in such studies under inflammatory conditions?
  2. Why the array experiments are not described in the methods section? Moreover, the authors may deposit their raw array data in a suitable repository.
  3. In the introduction section, mentioning a few recently published reviews on the role of microRNAs in disease, could be useful for interested readers. For example (PMID: 39938625, PMID: 40082357).

Round 2

Reviewer 1 Report

Comments and Suggestions for Authors

The authors have addressed my comments properly.

Reviewer 2 Report

Comments and Suggestions for Authors

The quality of the manuscript has been improved  and I endorse its publication